# Ex Vivo Culture Models to Indicate Therapy Response in Head and Neck Squamous Cell Carcinoma

**DOI:** 10.3390/cells9112527

**Published:** 2020-11-23

**Authors:** Imke Demers, Johan Donkers, Bernd Kremer, Ernst Jan Speel

**Affiliations:** 1Department of Pathology, GROW-school for Oncology and Development Biology, Maastricht University Medical Centre, PO Box 5800, 6202 AZ Maastricht, The Netherlands; imke.demers@mumc.nl; 2Department of Otorhinolaryngology, Head and Neck Surgery, GROW-School for Oncology and Development Biology, Maastricht University Medical Centre, PO Box 5800, 6202 AZ Maastricht, The Netherlands; j.donkers@student.maastrichtuniversity.nl (J.D.); bernd.kremer@mumc.nl (B.K.)

**Keywords:** head and neck cancer, primary cell culture, 3D cell culture, personalized therapy, preclinical prediction model, sensitivity testing, ex vivo model, drug response, organoid, histoculture

## Abstract

Head and neck squamous cell carcinoma (HNSCC) is characterized by a poor 5 year survival and varying response rates to both standard-of-care and new treatments. Despite advances in medicine and treatment methods, mortality rates have hardly decreased in recent decades. Reliable patient-derived tumor models offer the chance to predict therapy response in a personalized setting, thereby improving treatment efficacy by identifying the most appropriate treatment regimen for each patient. Furthermore, ex vivo tumor models enable testing of novel therapies before introduction in clinical practice. A literature search was performed to identify relevant literature describing three-dimensional ex vivo culture models of HNSCC to examine sensitivity to chemotherapy, radiotherapy, immunotherapy and targeted therapy. We provide a comprehensive overview of the currently used three-dimensional ex vivo culture models for HNSCC with their advantages and limitations, including culture success percentage and comparison to the original tumor. Furthermore, we evaluate the potential of these models to predict patient therapy response.

## 1. Introduction

Approximately 5.5 million people worldwide suffer from a form of head and neck squamous cell carcinoma (HNSCC). The yearly incidence rate is approximately 890,000 and 450,000 people die each year as a consequence of this disease [1,2]. This makes it the seventh most prevalent cancer type in the world with a 5 year survival rate of 25–60% depending on anatomical site and stage [3]. In addition to smoking and alcohol consumption, infections with high-risk human papillomavirus (HPV) are recognized as a risk factor for the development of oropharyngeal carcinomas, specifically [4]. In recent decades, large efforts in clinical care and research have been made in order to increase the 5 year survival rate. Cisplatin-based chemotherapy was introduced more than 40 years ago and is still regarded as one of the most influential adjuvant treatments for HNSCC. However, this treatment increases the 5 year survival rate by only 4%, illustrating the limited additional value of adjuvant treatments to date [5]. More recent promising progress in the treatment of HNSCCs is the development of targeted therapies and immunotherapy. The epidermal growth factor receptor (EGFR) inhibitor cetuximab was approved by the FDA in 2006 [6]. However, cetuximab was shown to be significantly more effective than standard treatment for HNSCC in only two situations—in combination with radiotherapy in patients with local progressive disease for whom chemotherapy is contraindicated, and in combination with platinum-containing chemotherapy in recurring or metastasized disease. Even though targeted therapies have given us an entirely new approach to treat HNSCCs, their impact on the 5 year survival rate is limited so far [7,8]. Much research is being conducted into specifically targeting other driver genes in oncogenic signaling pathways, such as mutations in the oncogene PIK3CA. This is achieved with the help of databases such as the Cancer Genome Atlas, which is the most comprehensive collection of integrated genomic annotations of molecular alterations in multiple cancer types [9,10,11].

Immunotherapy is the second new modality of treatment which has the potential to improve survival of HNSCC patients. Specifically, immune-checkpoint inhibitors are the subject of much attention. These drugs act by blocking inhibitory signals for T-cell activation, enabling tumor reactive T cells to overcome regulatory mechanisms and mount an effective anti-tumor response. At the moment, the most promising immune-checkpoint inhibitors are nivolumab and pembrolizumab, both inhibitors of the programmed cell death protein 1 (PD-1) receptor. In recurrent or metastasized platinum-resistant HNSCCs, these therapies increase the overall survival rate significantly when compared to standard treatment [12,13]. However, the overall response rate of HNSCC patients only seems to be up to 20% and the average overall survival time is increased by only a few months [14].

Not only the development of new therapies and bringing them to market maturity, but also the increasing need to test therapies in a personalized setting form a large challenge now and in the future. At this moment, new therapies for HNSCCs are mainly tested on a group level, which means that within a group of patients, different subgroups with different therapeutic and side-effects can be included. Therefore, it is difficult to predict the therapeutic benefit of a (new) treatment for the individual patient. Furthermore, newly developed (systemic) therapies are mostly tested in patients with the most progressed, usually palliative, stages of HNSCC, whose standard therapy has failed. Often, it is not known whether the same therapy has the same (side-) effects in other cancer stages. Therefore, it is desirable that a test method becomes available in clinical practice which allows for individual testing of a certain treatment during the diagnostic work up of the patient and allows for prediction of the therapeutic effect. This would be an important improvement in personalized medicine, which is not available in clinical practice yet.

Cell culture models offer the chance to fill this gap. In an optimal setting, tumor tissue can be cultured, and different therapies can be tested to predict therapeutic outcome before treatment of the patient. Because of the progression in the development of new therapies within the past 10 years, the interest in cell culture techniques increased as well. Multiple culture models have been developed and optimized for HNSCCs, with specific attention to cultures that grow in three-dimensional (3D) architecture. Because of the growing number of cell culture models, an overview of all these models with their (dis)advantages and purposes is required. Whereas Dohmen et al. published a narrative review in 2015 [15], there is a need for a more comprehensive update. While Dohmen et al. researched culture models in regard to chemotherapy (CT) or radiotherapy (RT) sensitivity testing, testing response to immunotherapies (IT) and targeted therapies (TT) has become more important since. Therefore, we review current literature on HNSCC primary 3D culture models and their application as preclinical predication assays for therapy response. 

## 2. Materials and Methods

A systematic literature search was performed using the PubMed and EMBASE databases. The search was built to include all articles discussing primary HNSCC 3D culturing techniques on which CT, RT, IT, and/or TT was tested (Appendix A). These articles were first screened based on title and abstract, after which a full-text screen of the selected articles was performed. Studies were included if they described a 3D culture technique of fresh primary HNSCC tissue in combination with sensitivity testing to aforementioned therapies. Studies describing culture models involving animals were excluded. Conference abstracts and reviews were also excluded, but their references were screened for additional articles. The titles and abstracts of the references of all included articles were screened as well (Figure 1).

Full tables with all 53 included articles and their extracted data can be found in the Appendix A. Of these, key publications were selected and presented in the results section. This selection was performed by two authors scoring all articles based on predefined criteria (Table 1). All articles scoring 5 or more points were defined as key publications. Studies describing a correlation between ex vivo and patient treatment response were selected and of all culture models at least the highest scoring article is presented since not every culture model was described by an article scoring at least 5 points. The screening of articles by titles and abstracts was performed by one author, whereas the final selection, data extraction and scoring of articles based on the full texts was performed by two authors independently. Final selection was based on the consensus of all authors.

## 3. Results

### 3.1. Overview of Ex Vivo Culture Models Used for HNSCC

Based upon the articles found in the systematic search and additional relevant literature, an overview of the most commonly used primary culture models for HNSCCs was composed. In this overview, we aim to give a description, nomenclature and (dis)advantages of each model (Table 2). For the remainder of this review, this terminology will be used to describe the culture models in the included studies. To make the overview as complete as possible, 2D monolayer and patient-derived xenograft (PDX) culture models were also included for comparison purposes, even though these were excluded in the systematic search. 

#### 3.1.1. Adherent Monolayer

In order to establish a 2D monolayer culture, the tumor sample is dissociated into single cells and cultured at the bottom of a container, such as a culture flask or Petri dish (Figure 2 and Figure 3A). Due to clonal expansion, the cells will cover the entire surface. Synthetic culture medium, often supplemented with fetal bovine serum and L-glutamine, is used to provide cells with nutrients and growth factors. Cell culturing is usually performed at body temperature (37 °C) and a subculture is achieved by detaching the cells from the plastic surface with trypsin and/or EDTA when the cells reach confluency. Since the technique is relatively inexpensive, well established and relatively easy to perform, it remains one of the most used culture techniques in the world [44]. However, when culturing cells as a monolayer, the original tissue preservation is lost. This causes changes in cell morphology and interactions. In addition, as monolayer cultures are usually formed due to clonal expansion, selection for one cell type often takes place. This results in a monoclonal culture that may also change phenotypically and genotypically over time [45,46]. This monoclonality is in contrast to the original tissue containing multiple cell types. These discrepancies between monolayer cultures and the in vivo situation have caused the scientific community to start developing 3D culture techniques, which eliminate some of these shortcomings.

#### 3.1.2. Three-Dimensional Culture Models

Three-dimensional culture models have become more popular in recent years because they mimic the tumor architecture inside the body more accurately compared to monolayer cultures. In general, 3D models provide a more realistic way to grow tumor cells, including a better imitation of the variable access to nutrients and oxygen, enabling assessment of the tissue-penetrating ability of drugs and allowing for interaction between different cell types. All 3D culture models have their own unique advantages, disadvantages and applications. As there is a lot of confusing and overlapping terminology for the 3D culture models, especially for sphere-type models, this overview aims to clarify this nomenclature.

#### 3.1.3. Multicellular Spheroids

Multicellular spheroids are cell aggregates from single-cell suspensions or tissue fragments containing multiple cell types from primary tissue (Figure 2 and Figure 4A). To achieve cell aggregation, a multitude of methods are available including suspension cultures with ultra-low attachment plates or hanging-drop cultures (Figure 3C–G). For this purpose, primary tissue can either be enzymatically dissociated into single cells or it can be only mechanically minced into small fragments [22]. Spheroids formed by the latter are also referred to as organotypic multicellular spheroids or fragment spheroids. These spheroids leave part of the original microenvironment of the original tissue intact as the cells are not dissociated from their original environment [20].

#### 3.1.4. Cancer Stem Cell-Enriched Spheroids

The cancer stem cell (CSC)-enriched spheroids (also referred to as tumorspheres or tumor-derived spheroids in literature) originate from CSC or cells with stem cell traits. The enrichment for these CSCs is often performed by cell sorting (i.e., based on CD44 expression) and assessment of self-renewal capability. These CSCs are grown in low-adherent conditions using stem cell medium in order to form spherical structures [47]. Whereas cell aggregation may occur by the low-adherent conditions, CSC-enriched spheroids are predominantly formed by clonal expansion of the CSCs (Figure 4B). The self-renewal capacity of the CSCs gives these spheroids the potential to proliferate and differentiate, which is of importance when studying stem cell characteristics and behavior. In contrast to the multicellular spheroids, CSC-enriched spheroids generally contain only one cell type (monoclonal) and therefore differ substantially from the original tumor on a histological level. It also remains a challenge to correctly identify the cells that could be denominated as CSCs, which is currently only possible by evaluation of stem cell-specific markers or self-renewal capacity [47].

#### 3.1.5. Organoids

Already in 1992, organoids from disaggregated carcinomas were established in athymic mice by Köpf-Maier and colleagues [28]. The first organoid cultures for HNSCC without the use of animals were described in 2018 [30,48]. Organoids develop from stem cells or organ progenitors but contain multiple types of organ/tumor-specific cells through lineage-dependent differentiation (Figure 4C). This is achieved by embedding the fresh primary dissociated tumor cells in an extracellular matrix (ECM), such as Matrigel, and providing the cultures with specifically supplemented growth medium (Figure 3D). This causes the organoid to self-organize in a manner similar to the in vivo situation. Because the organoid culture technique is relatively new and generally complex, it still faces practical challenges, including the fact that it is time consuming, high costs and the need for well-established protocols which differ depending on the tumor type.

#### 3.1.6. Histocultures

Histocultures consist of tissue that has only been modified mechanically without enzymatic dissociation (Figure 2). This culture model is described in the literature as tumor slices, tumor fragments, tumor particles, (ex vivo) tumor explants or tumor sections. Histocultures preserve the tumor cells in their original microenvironment, including the ECM and immune, stromal and vascular cells. The tumor cultures are generally cultured at air–medium interface, using, e.g., culture inserts (Figure 3H). The major challenge of culturing histocultures is the quick deterioration of the tissue and loss of cell viability [36]. This makes them not suitable for culturing for extended periods of time at this moment. Tissue viability might be prolonged by the use of a supportive matrix [49]. Nevertheless, the histoculture model is hampered by a relatively short viable period of 3–6 days and other disadvantages such as low throughput and low reproducibility.

#### 3.1.7. Patient-Derived Xenograft (PDX)

A different way of growing 3D tumor models is to implant the patient-derived tumor cells or tumor fragments into immune-deficient mice (Figure 3I). This allows outgrowth of the tumor cells and thereby retaining the intratumor heterogeneity [50]. Evaluation of tumor growth within mice offers the chance to evaluate tumor formation in a living system, which allows investigation of metastatic processes and the influence of the endocrine system. Nevertheless, the culture technique does have major flaws. The most fundamental flaw is that the tumor and its microenvironment are slowly mingled with ECM and cells from the mouse, which will likely influence the test results. In addition, the PDX cultures usually have a long generation time (approximately 2–12 months) and are fairly expensive due to maintenance of animals and their facilities [51]. In addition, PDX cultures are unsuited to assess the role of the immune system in relation to therapy response due to the immune-deficient nature of the mice. Lastly, ethical issues are involved with the use of animals in (cancer) research. These include using the minimum number of animals required, allowing precise statistical analysis and results, and preventing the repetition of experiments. Another ethical concern is the physical and moral well-being of the animals, for which efforts should be undertaken to replace, reduce, and refine animal experiments (three Rs principle) [52,53]. For these reasons, we decided not to evaluate PDX models as a potential preclinical model in the current review.

#### 3.1.8. Microdevices

Microdevices are unique as they can be used to culture multiple monolayer and 3D models (Figure 3B) [54]. Therefore, overlap may be observed between microdevice cultures and the aforementioned culture models, e.g., regarding culture success rate. Microdevices allow a controlled culture-environment, including continuous perfusion of the culture with medium, mimicking constant blood flow in the in vivo situation. In addition, a microdevice can offer special structures to control the position, shape, function and both chemical and physical cell environments [55]. The main drawback of using a microdevice in the culture setup is that all these factors increase the complexity substantially [56]. Besides that, costs are still high for the setup of a microdevice system and read-out methods are limited.

### 3.2. Characteristics of Primary 3D Culture Models of HNSCCs and Suitability for Drug Response Testing

Key publications describing the use of primary 3D cultures of HNSCC for sensitivity testing to CT/RT and IT/TT are presented in Table 3 and Table 4, respectively. For each study, model characteristics, technical aspects such as culture duration and success percentage, and main results are presented. Table 3 includes an overview of the correlation between therapy response observed ex vivo compared to the clinical response of the patient with predictive values including sensitivity and specificity. For IT/TT (Table 4), this correlation has not been reported in any of the reviewed articles. The culture models are grouped by technique in chronological order. Full tables with all examined articles can be found in the Appendix A. Below, studies using primary culture models for drug sensitivity testing are discussed per culture model.

#### 3.2.1. Multicellular Spheroids

Four out of seven studies using multicellular spheroids reported success percentages of 50–100%, >90% and two times 100%. The tumors originated from different HNSCC locations, including oropharynx, hypopharynx, larynx, tongue, and unknown primary site. In regard to culture success rate, one study reported that spheroid formation with primary cells obtained from biopsies was more reliable and reproducible in ultra-low attachment plates than in a hanging-drop system [22]. The range of culture duration of these spheroids was 4–21 days, with an average of 10–15 days.

##### CT/RT

One study used a multicellular spheroid model of HNSCC for cisplatin, 5-FU, and radiotherapy sensitivity testing [57]. This study analyzed aldehyde dehydrogenase (ALD)-positive and ALD-negative subpopulations in these spheroids and examined ALD activity compared to primary monolayer cell cultures. Spheroid cultures show 1–2% apoptosis after treatment, in comparison with 5–25% in 2D monolayer cultures. This observation indicates differences in response to drugs between 2D and 3D culture models and suggests that the 3D architecture might be a better representation of the tumor in vivo.

##### IT/TT

Three studies of Heimdal and Olsnes describe multicellular spheroids in co-culture with monocytes or monocyte-derived macrophages [20,77,78]. To elucidate the mechanisms of monocyte cytokine secretion, fragment spheroids (F-spheroids) from malignant and benign mucosal tissue were cultured in the presence of monoclonal antibodies against CD14, CD29, and MCP-1, molecules involved in monocyte activation and infiltration. Tumor samples from a total of 24 patients were investigated. The monoclonal antibodies affected cytokine secretion, including MCP-1, IL-6, and TNF-a, but the effect on cancer cell viability or survival have not been investigated. However, the same group showed in a separate study that increased levels of IL-6 in these co-cultures are predictive for disease recurrence in HNSCC patients [79].

F-spheroids have also been used in a subsequent study of Kross et al. [70]. The main goal was to analyze tumor-associated macrophage cytokine secretion by treating the spheroids with L-leucine-methylester (LLME), a drug which selectively induces apoptosis in macrophages, but not in tumor cells. LLME treatment only affected the macrophages and their cytokine secretion, without influencing the viability of the tumor cells within the F-spheroids.

Another study using co-cultures was conducted by the same group [71]. Tissue from five patients was used to co-culture multicellular spheroids with Natural Killer (NK) cells. They determined cytotoxic activity of the NK cells after pre-treatment of the spheroids with cetuximab. NK cells showed clearly improved and more organized function when cetuximab was added, which resulted in a higher percentage of killed tumor cells. This observation supports the suitability of this co-culture model to evaluate treatment response.

#### 3.2.2. CSC-Enriched Spheroids

Two out of five studies using CSC-enriched spheroids reported success percentages of 6% and 80–100% [24,26]. The amount of time required for the cultures varies between 6 and 17 days, with an average of 12 days.

##### CT/RT

The first sensitivity testing with CSC-enriched spheroids was conducted by Lim at al., investigating culture response to cisplatin, 5-FU, paclitaxel and docetaxel [24,25]. It was observed that undifferentiated spheroids were more chemo-resistant than differentiated spheroids. As an explanation, they showed that undifferentiated spheroids consisted of 1.74% extra chemo-resistant cells, while this percentage was only 0.11% in differentiated spheroids. A second study confirmed this finding by showing that stem cells grown as spheroids or as an adherent monolayer were relatively more chemo-resistant compared to the same culture models consisting of differentiated cells [25]. The CSC-enriched culture model is interesting to investigate stem cell behavior and characteristics, but the observed differences in drug response in relation to differentiation state might question whether CSC-enriched spheroids are a representative model for the in vivo situation.

A subsequent study investigated radio-sensitivity and migratory potential of CSC-enriched spheroids derived from five patients [26]. They observed no statistically significant difference in surviving fraction and spheroid migration after treatment with radiation doses up to 10 Gy, compared to the untreated control. This is in line with the findings of the previous studies reporting on the chemo-resistance of CSCs.

##### IT/TT

To overcome therapy resistance, therapies specifically targeting CSCs in HNSCC are also explored with the use of CSC-enriched spheroids. One study investigated therapeutic inhibition of c-Met, which is identified as a self-renewal marker of CSCs in HNSCC patient-derived tumor xenografts [80]. They showed that CSCs were indeed more sensitive to c-Met inhibitor PF-2341066 than to docetaxel, whereas differentiated cells showed the opposite response [72].

#### 3.2.3. Organoids

The organoid culture technique is relatively new and has not been extensively investigated for HNSCC yet. Reported success percentages vary from 30.2% to 80%. It is reported that organoids can be established in up to 7 days but may be kept in culture for prolonged time if required [30,31]. Drug testing or passaging is recommended after 10–14 days culturing [30,31,48,81].

##### CT/RT

One study investigated response to cisplatin, docetaxel and 5-FU and reported IC50 values for several organoid lines. These organoids showed similar histological patterns and expression levels of vimentin and stem cell markers CD44 and ALDH1A1 when compared to their original tumors [30]. IC50 values from organoid drug treatment in vitro were observed to be similar to the drug response in vivo, after injecting these organoids in mice. Interestingly, another study observed that the successful formation of organoids was significantly associated with poor response to presurgical neoadjuvant chemotherapy or chemoradiation in their patients. In addition, IC50 values for 5-FU of the organoids were much higher for organoids after passaging (0.4–1.4 µM vs. 23.6–53.6 µM), which is attributed to an increased CD44 expression and autophagy [48].

One year later, the Clevers group published two studies using HNSCC organoids with an extensive description of methods and organoid characterization [31,81]. When comparing organoids with the original tumor they observed that specific histopathologic changes were retained in culture. However, the organoids only contained the transformed epithelial tumor cells and not the connective tissue, immune or vessel elements. Drug screens were performed on the organoids and IC50 values were reported for cisplatin and carboplatin, showing differential sensitivity of the organoids to these compounds. Area under the curve (AUC) values were calculated for radiotherapy treatment and compared to clinical response of the patients who received (postoperative) radiotherapy. Interestingly, six out of seven patient outcomes matched with the responsiveness of their respective organoids. The organoid of the seventh patient showed to be resistant to radiotherapy in the in vitro assay, whereas the patient showed no signs of recurrence five months after treatment. Longer follow up should reveal whether this patient relapses in the coming months. Even though this is a small population size, this result shows potential for the use of organoid cultures to predict individual radiotherapy response.

##### IT/TT

On the basis of mutations detected in their organoids or in HNSCC in general, the same study also tested organoid sensitivity to several targeted therapies, including cetuximab, nutlin-3 (p53-MDM2 inhibitor), alpelisib (PIK3CA inhibitor), vemurafenib (BRAF inhibitor), everolimus (mTOR inhibitor), AZD4547 (FGFR inhibitor), and niraparib (PARP inhibitor) [31]. No correlation between EGFR expression and cetuximab response was observed. However, organoids insensitive to cetuximab often carried mutations downstream of EGFR. Increased sensitivity to vemurafenib was observed in a BRAF-mutant organoid line, but no correlation was found between responsiveness to alpelisib and PIK3CA mutations. Although mutations in PARP, mTOR and FGFR were not detected in the organoid lines, variable sensitivities to these compounds were observed.

#### 3.2.4. Histocultures

The success percentage of the histoculture model varies from 59% to 100%, with an average of 90% and a median of 98%, as reported by 21 out of 29 articles. Studies describe a culture duration 1 up to 21 days [32,49,82]. In general, the average culture duration of histocultures was 5 days, with a median of 5 days.

##### CT/RT

The first two groups reporting on HNSCC histocultures determined sensitivity to cisplatin, 5-FU and mitomycin C treatment (Table 3). They observed that viable regions of the cultures were histologically very similar to the original tumor, although regions with necrotic tumor tissue were observed [32,58,59]. The authors presented IC50 values and all three compounds were able to decrease the ^3^H-TdR incorporation in different histocultures. Hasegawa et al. showed that cisplatin and 5-FU were also able to decrease cell viability by an MTT read-out method [64]. In 2013, Gerlach et al. cultured tumor sections on a membrane culture insert [34]. They also observed that the cultures were viable and maintained their typical morphological features in vitro for up to 6 days when compared to the original tumor. DNA double strand breaks and cell proliferation was assessed by γH2AX and Ki-67 expression, respectively. Untreated cultures were found to maintain a high proliferative activity and no change in DNA damage was observed over time. Treatment with cisplatin and docetaxel resulted in apoptotic fragmentation, activation of the apoptosis marker caspase-3, and cell loss within the histocultures [34].

The first comparison between cisplatin response in culture and in vivo was later performed by one of these groups [32]. In this study, they presented predictive data on sensitivity (71%), specificity (78%), positive predictive value (PPV) (83%) and negative predictive value (NPV) (64%) (Table 3). More studies followed, investigating clinical correlation with multiple types of chemotherapy and larger patient groups [60,62,63,65]. Generally, these studies reported a good correlation between ex vivo response and clinical response. Whereas the overall sensitivity was relatively high (79–91%), two studies showed a specificity of approximately 50% [62,63]. One of these studies reported that 17 out of 19 patients tested for individual drugs in vitro received a combination of chemotherapies and even in combination with radiotherapy [62]. In a subsequent study, 97% of the patients received the same drug or combinations of drugs that was studied in vitro, making the interpretation of the clinical correlation more reliable [65].

In addition to clinical drug response, two studies investigated the correlation between in vitro drug response and patient survival. A significantly greater 2 year cause-specific survival was described when ex vivo cultures were sensitive to 5-FU and cisplatin [61]. In line with this, another study showed that a high efficacy of cisplatin in vitro (Inhibition Index > 50) was significantly correlated with a better overall survival [66].

The most recent study on HNSCC histocultures by Engelmann et al. described the longest culture duration so far [49]. With the use of a dermal equivalent (DE), they were able to maintain tumor explants of all their non-HPV-driven HNSCCs up to 21 days in vitro. This DE was composed of healthy human-derived fibroblasts and viscose fibers and served as a scaffold for the tumor sample. The authors could distinguish three growth patterns, including an invasive pattern, showing scattered irregular clusters of tumor cells invading the DE, an expansive growth pattern, showing horizontal tumor cell spreading on top of the DE, and a silent growth pattern, without invasion or horizontal spreading. Treatment of the cultures with radiotherapy showed variable responses characterized by expression levels of apoptosis (caspase-3-positive cells). Two out of five irradiated samples showed an increase in caspase-3 expression, with both of these samples being HPV driven. Interestingly, one patient developed local relapse 17 months after surgery and radiotherapy, with the corresponding ex vivo culture showing an invasive growth pattern. Unfortunately, sensitivity to ex vivo radiotherapy was not examined on this tumor sample. Importantly, this study described that culturing HPV-driven tumor samples appeared to be more challenging compared to non-HPV-driven tumors. Although they were able to maintain HPV-driven samples for up to 21 days, they observed that half of these cultures showed either decreased levels of p16 or decreased amount of cancer cells on day 14.

##### IT/TT

Dean et al. were the first to report the use of HNSCC histocultures for IT/TT sensitivity testing in 2010 [73]. They performed sensitivity testing for cetuximab and a monoclonal antibody against extracellular matrix metalloproteinase inducer (EMMPRIN), a cell surface molecule known to promote tumor growth and angiogenesis in HNSCC. It was observed that tumor sections were viable for up to 72 h and that less than 5% of the specimens showed necrosis. Anti-EMMPRIN therapy resulted in a reduced cell proliferation and an increase in caspase-mediated apoptosis. In addition, a larger percentage of ex vivo cultures was sensitive to the anti-EMMPRIM antibody compared to cetuximab (58% vs. 33%).

Sensitivity testing to cetuximab was investigated by four studies [34,36,73,76]. Three of these studies used smaller tumor slices than described previously (300–350 µm in thickness). Concerning ex vivo tissue viability, contradictory results were presented by these groups. As mentioned earlier, Gerlach and colleagues reported a good tissue viability with a high proliferative activity for up to 6 days, whereas another study observed a 30–70% decrease in cell proliferation after 48 h and after 72 h necrosis has increased significantly without treatment. This resulted in an average of 25% proliferating (Ki-67-positive) cells in the control samples [36]. In general, it was observed that cetuximab decreased cell viability (ATP levels), the number of nuclei, and number of Ki-67-positive cells, while the number of apoptotic (caspase-3-positive) cells was increased.

In addition to cetuximab, other targeted therapies that are not used in clinical practice have been tested on HNSCC histocultures. It was presented that treatment with the PI3K inhibitor LY294002 sensitizes ex vivo cultures to radiotherapy, resulting in increased DNA damage and decreased cell proliferation [35]. No reduction in cell proliferation was observed after treatment of histocultures with the RAF kinase inhibitor sorafenib [36]. Lupeol, a naturally occurring phytochemical found in fruits, vegetables and plants was also tested for its effects on cell viability and proliferation. Lupeol treatment showed profound decrease in proliferation (Ki-67 expression) compared to control tissues [74]. Ex vivo treatment with the MEK inhibitor PD-0325901, either in combination with radiotherapy or as monotherapy, only showed modest effects on cell proliferation. This might be attributable to the very low proliferation fraction in control tissues in this study (5% to 7.5%). MEK inhibition prior to irradiation decreased p-ERK levels and increased γH2AX levels predominantly in one patient sample with low basal γH2AX expression [75].

Donnadieu et al. cultured tumor slices of HNSCC and exposed them to a panel of targeted therapies [76]. These therapies were selected based on their inhibitory effect on oncogenic kinases and reached phase II/III in clinical trials for the treatment of various solid tumors, including EGFR, B-RAF, KIT, HGFR, FRFR, and mTOR. They observed that effect of treatment varied depending on drug and patient. The multi-kinase inhibitor sorafenib proved to be most effective in inhibiting cell proliferation (5/14 tumors). In total, a more than 50% inhibition of proliferation was observed in 10/14 tumor samples for at least one drug. Although the levels of ERK and p-ERK were determined, no mutational analysis of these oncogenic pathways was described on the ex vivo cultures, which could correlate to drug response.

#### 3.2.5. Microdevices

Whereas microdevices could be designed for the maintenance of various ex vivo culture models, current literature on HNSCC often describes the use for these devices to study histocultures. Success percentages of 67%, 91% and 100% are reported with culture durations varying from 2 to 10 days. One study compared four different culture models with the use of microdevices, including a monolayer, spheroid and histoculture model [54]. This comparison showed the importance of stable culture conditions and revealed that the choice of cell culture format might play a role in the physiology of the cultured cells and outcome of drug sensitivity assays.

##### CT/RT

Hattersley et al. were the first to report on culturing HNSCC tissue with the use of a microdevice and tested sensitivity to cisplatin and 5-FU [41]. The nuclei of the tissue seemed intact after 72 h, and the percentage of viable cells after 7 days was 72% in the control samples. They specified that there was no evidence of central necrosis, which could be attributed to the microfluidic diffusion. A decrease in cell viability (decrease in WST-1 metabolism, increase in LDH release) and induction of apoptosis (increased cytochrome-c release) were observed after treatment with both compounds.

The efficacy of radiotherapy on HNSCC histocultures is also investigated with the use of microdevices [67,68]. The first study observed a significant increase in cell death, measured by LDH release, 2 h after irradiation of the tissues with 40 Gy [67]. Whereas there was no difference in apoptotic activity (<2%) between control and uncultured tumor samples, a dose-dependent increase in apoptosis was observed in the radiotherapy treated tissues. In line with this, a second study detected increased apoptosis and higher levels of DNA fragmentation after irradiation. Expression of γH2AX was raised after treatment, but not significantly. The percentage of proliferating cells decreased in a dose-dependent way following irradiation. In the same study, the correlation of ex vivo radiosensitivity and clinical response is also investigated. Although clinical information was only available for two patients, matched responses were observed for both patients and their representative ex vivo cultures. Important to mention is that this study used four markers to predict response to radiotherapy in vitro (LDH release, γH2AX expression, CK18-LI, DNA fragmentation, and Ki-67 expression), and for each patient, only two of these markers were matching clinical response. In addition, one of the patients received chemoradiotherapy, whereas only radiosensitivity was determined ex vivo [68].

A recent study determined radiosensitivity in combination with cisplatin [69]. They observed that γH2AX expression and the number of apoptotic cells were similar in untreated control and pre-culture samples, whereas the cell proliferation (Ki-67) had decreased in control samples when compared to the pre-culture samples. Irradiation reduced proliferation (BrdU), increased DNA damage (γH2AX), and caspase-dependent apoptosis (caspase-cleaved cytokeratin-18). Caspase-dependent apoptosis was further increased by concurrent cisplatin treatment.

##### IT/TT

Microdevices are also used as a co-culture system with immune cells to examine immune cell migration and cancer cell proliferation in response to an PDL-1 antibody and IDO 1 inhibitor [42]. This study showed that IDO 1 inhibitor, but not PD-L1 inhibitor, induced immune cell migration towards cancer cells. Drug efficacy on cell proliferation was variable between the two tumor samples from HNSCC patients. Since immune cell migration did not parallel the effect on cancer cell proliferation, it is considered that immune cell migration is not sufficient to evaluate therapy response to immunotherapeutic drugs in this setting.

## 4. Discussion

With this review, we aim to provide a comprehensive overview of the current literature on ex vivo 3D culture techniques for HNSCC and evaluate their suitability as a preclinical prediction assay for individualized therapy selection. With the increasing knowledge on driver mutations and deregulated cellular pathways in HNSCC, the development of new (targeted) treatments, and the varying response rates for both standard-of-care and new therapies, a reliable prediction assay for therapy response is more important than ever.

When culturing primary tissue or cells in general, multiple aspects need to be considered in relation to culture success percentage. It is essential to minimize the time between surgery and start of the culture, since cutting of the blood supply (ischemia) could lead to fast deterioration of the tissue [83]. Furthermore, primary cells have a limited lifespan and are more sensitive to environmental changes and stress compared to immortalized cell lines. In addition, primary cultures are prone to microbial contaminations with bacteria and/or fungi, especially when the tissue is derived from locations with an extensive microbiome, such as the intestines or oral cavity [84,85]. Contamination with fibroblasts could also be a practical challenge, since fibroblasts are able to overgrow the culture because of their high proliferation rate [86,87]. For primary cultures of HNSCC specifically, it remains a challenge to successfully culture and maintain HPV-positive cells and tissues in vitro [49,88,89]. The exact explanation is still unknown, but it is thought that tumor cells must have acquired traits or mutations compatible with survival and immortality to be able to survive in the unnatural in vitro environment. This is supported by the fact that almost all currently used HPV-positive HNSCC cell lines are from smoking patients with aggressive tumors that fail to respond to initial therapy [88,90]. Furthermore, the stromal microenvironment is thought to be involved, if not essential, in HPV-positive epithelial cell growth and disease initiation and maintenance by reciprocal epithelial-stromal interactions [91,92]. Thus, HPV-positive tumor cells might require the presence of (specific factors within) the microenvironment in order to survive in vitro. Recent data have shown that HPV-positive tumors are a heterogeneous group and can be subclassified based on genomic profiles (e.g., characterized by a signature of mesenchymal and immunological response genes (HPV-IMU), or keratinocyte differentiation and oxidative stress genes (HPV-KRT), with the latter subgroup showing more frequently integrated HPV and enrichment of PI3KCA mutations), EGFR expression, and HPV integration status, amongst others [93,94,95]. In addition to patient prognosis, these factors might also influence in vitro viability of HPV-positive tissues. In the investigated literature, specific information on virus positivity of the tumor in relation to ex vivo culture success rate is limited.

One of the essential requirements for a reliable tumor model is the resemblance to the original tumor composition as closely as possible, since the tumor-microenvironment, including multiple cell types and tumor-stroma interactions, has shown to influence tumor behavior and therapeutic response [96,97]. In addition, culture success rate and culture duration are important aspects for a tumor model to serve as a preclinical prediction assay. In this review, we show that the best culture success rates have been achieved with the histoculture technique. Furthermore, as this culture method does not require enzymatic dissociation, natural tumor heterogeneity, cell–cell interactions and cell-stroma interactions are left intact, resulting in the best simulation of the in vivo situation as possible. An additional advantage might be the relatively short-term culture duration of this tumor model which reduces the chance on phenotypic and genetic alterations, as observed in more long-term cultures, allowing for fast decision making in a personalized therapy approach. In contrast, the often relatively quick occurrence of tissue deterioration during culturing might influence the outcome of drug sensitivity assays [36]. Improving tissue viability over time, for example by the use of a dermal equivalent (consisting of viscose fibers and human-derived fibroblasts) as tissue support, could increase reliability of the histoculture model and allow for prolonged ex vivo drug exposure [49]. In addition, microdevices might offer a chance to increase tissue viability by providing a controlled culture environment and continuous perfusion and nutrient supply to the tumor tissue. However, there is no convincing evidence yet for the role of microdevices in prolonging HNSCC tissue viability compared to conventional culture methods.

Longer culture durations are reported for HNSCC spheroid and organoid models evaluated in this review. Organoids specifically can be expanded for a long period of time and cryopreserved, which allows for a wide range of research applications, such as genetic modification and a prolonged exposure to anti-cancer drugs [98]. In addition, less tumor material is required for organoid generation, compared to histocultures. However, organoids only comprise of (transformed) epithelial cells, without native micro-environment with stromal compartment, immune cells, nervous system, and vessel elements [31,99]. The possibility to co-culture these models might offer an opportunity to overcome this limitation and study the interaction between different cell types. The same limitation is observed for spheroid models, especially when cultures are enriched for CSCs. Whereas this model could be interesting to investigate stem cell characteristics and behavior in relation to drug resistance, the resemblance to the original tumor might be questioned. In addition, identification and isolation of stem cells from tissues remains a challenge and is often based on stem cell markers, such as CD44 and ALDH. However, none of these stem cell markers has proven to identify CSCs with adequate sensitivity and specificity. Besides this, there are other unresolved aspects, such as the impure and variable stem cell population in human tumors and the stability of CSC immunophenotype over time [100,101]. Whereas there is a selection for cell type in organoids and CSC-enriched spheroids, the organotypic multicellular spheroids (or F-spheroids) are established by only mechanically modification of tissues, similar to histocultures. In contrast, these F-spheroids were cultured to form rounded spheres before use in sensitivity assays, which took typically 10–14 days [20,70,77,78]. During this period of spheroid generation, loss of epithelial cells from 28% to 12.9% was observed by one of these groups in a separate study [102].

In addition to aforementioned technical considerations, a preclinical prediction assay should be able to accurately predict patient therapy response. So far, clinical correlations were mainly reported by studies investigating sensitivity to chemotherapy with the use of ex vivo histocultures. Overall, this technique shows good predictive values (accuracy of 74–79%). However, two out of four studies describe a specificity of approximately 50% [62,63]. This means that half of the cultures were sensitive to chemotherapy ex vivo, while the corresponding patient showed no clinical response. This might be explained by mechanisms of resistance in vivo in addition to those at cellular level, for example the variation in pharmacokinetics between different patients. If the tumor cells are highly resistant ex vivo, there is a small chance that the drug will be effective in vivo. Therefore, it is argued that these (chemo)sensitivity assays might be a better predictor for therapy resistance than sensitivity [103]. Whereas increased tumor response rates do not necessarily increase patient survival, evidence is needed from clinical trials investigating patient survival in correlation to ex vivo drug sensitivity. Two studies investigated this correlation and reported a better cause-specific survival and overall survival when histocultures were sensitive to chemotherapy [61,66]. However, only chemosensitivity was tested ex vivo, while patients in both studies often received a combination of treatments with radiotherapy and/or surgery, which might cause a bias in survival data.

Evidence for the predictive value of organoid and spheroid models of HNSCC is still sparse. One recent study showed a correlation between organoid radiosensitivity and clinical responses in six out of seven patients [31]. Although the number of patients is small and these tumors comprise a heterogeneous group, all patients were treated with (postoperative) radiotherapy only. This allows for a reliable comparison between ex vivo and patient response. An ongoing study of the same group aims to include approximately 80 patients to follow up on these initial findings and elucidate whether organoid responses hold predictive potential for patient responses.

Although immunotherapy has become a new promising treatment modality for HNSCC with varying response rates, ex vivo sensitivity to these therapies has not been correlated to clinical response in the reviewed literature. Multiple studies do show the possibility to maintain and include immune cells in culture, which is essential to assess immunotherapy sensitivity [42,70,71]. With the exception of cetuximab, therapies targeting specific mutations are not routinely used to treat HNSCC patients yet. This makes it difficult to correlate ex vivo findings to the clinical situation. Nevertheless, it is of interest to investigate mutation status of the tumor in correlation to ex vivo sensitivity to targeted therapies [31,104]. In this context, Driehuis et al. observed increased sensitivity to a BRAF inhibitor in a BRAF-mutant organoid line derived from a BRAF-mutant HNSCC. In other cases, no correlation was observed, for example between organoid EGFR expression and cetuximab response and the presence of PIK3CA mutations and the responsiveness to PI3K inhibitor alpelisib [31].

As ex vivo cell culture models have matured in recent years, they have not become part of clinical routine yet. For this purpose, efforts should be made to improve technical aspects of all culture models in order to more closely resemble the original tumor (-environment), increase ex vivo cell viability and culture success rates, also for HPV-positive tumors. The presence of immune components in culture is not only essential for evaluating immunotherapy sensitivity, but may also influence sensitivity to other therapies, including chemotherapy, radiotherapy and targeted therapies [105,106,107]. In addition, larger studies should focus on obtaining more evidence on the predictive potential of ex vivo models with both tumor response and patient survival, in which ex vivo and in vivo treatment should be similar to allow for a reliable comparison and prediction. In addition, the application of testing targeted therapies would be most interesting for those tumor subtypes that require additional treatment or are characterized by an unfavorable prognosis, for example caused by radioresistance. Lastly, the use of unambiguous terminology should be a prerequisite for all studies reporting on 3D culture techniques. This will ensure that evaluating and comparing future research as well as working towards the best preclinical prediction model will be improved.

## 5. Conclusions

There is a strong need for preclinical 3D models of HNSCC, which allow prediction of therapeutic response in a personalized setting and furthermore enable novel drug testing before introduction into clinical practice. In this review, we observed that a wide range of ex vivo culture techniques have been introduced for HNSCC, all with their own advantages, limitations, and applications. So far, most information is available on HNSCC histocultures and their use to obtain an indication for response to chemotherapy. Future research should elucidate whether histocultures and/or other ex vivo tumor models can mature further to useful clinical tools.

## Figures and Tables

**Figure 1 cells-09-02527-f001:**
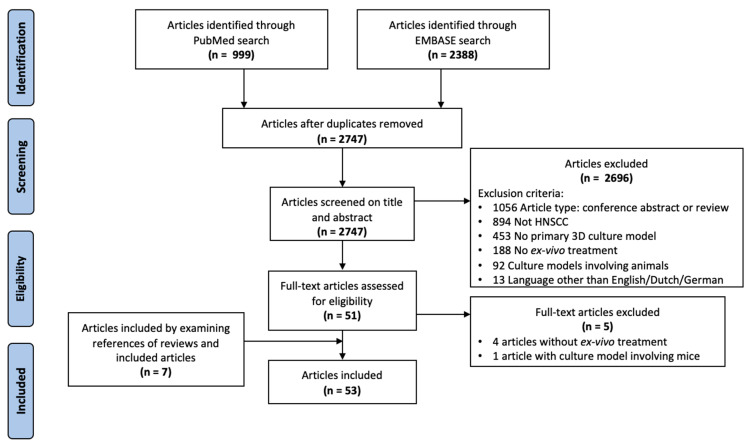
Flow diagram of the systematic literature search performed.

**Figure 2 cells-09-02527-f002:**
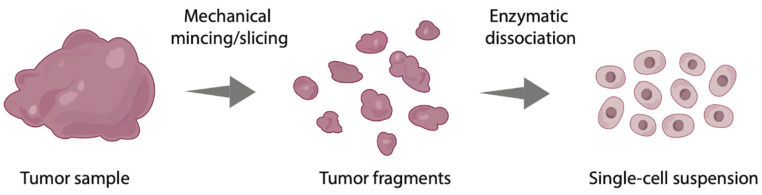
Processing of the primary tumor sample into tumor fragments by mechanical modification and subsequently into a single-cell suspension by enzymatic dissociation.

**Figure 3 cells-09-02527-f003:**
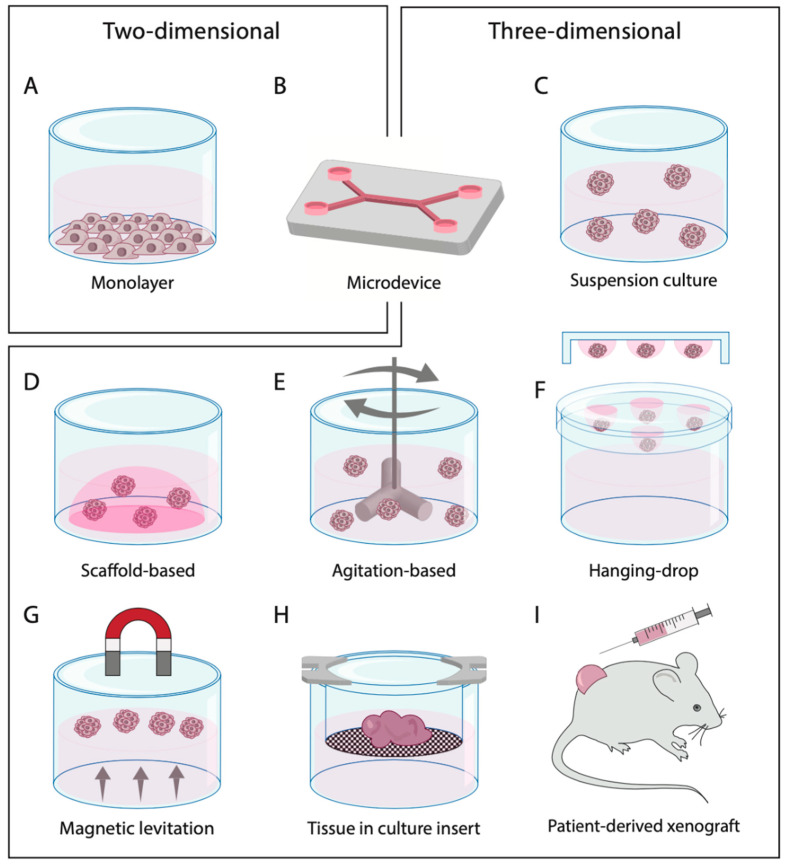
Primary cell culture techniques divided into two-dimensional and three-dimensional models. (**A**) 2D monolayer from single-cell suspension; (**B**) micro(fluidic) device; (**C**) spheroids in suspension culture; (**D**) spheroids embedded in a scaffold-based system; (**E**) spheroids in an agitation-based system, e.g., a spinner flask; (**F**) spheroids in hanging-drop cultures; (**G**) spheroids formed by magnetic levitation; (**H**) histoculture in culture insert; (**I**) patient-derived xenograft mouse model with subcutaneous injection. Image of xenograft was modified from Servier Medical Art (http://smart.servier.com/), licensed under a Creative Common Attribution 3.0 Unported License (https://creativecommons.org/licences/by/3.0/).

**Figure 4 cells-09-02527-f004:**
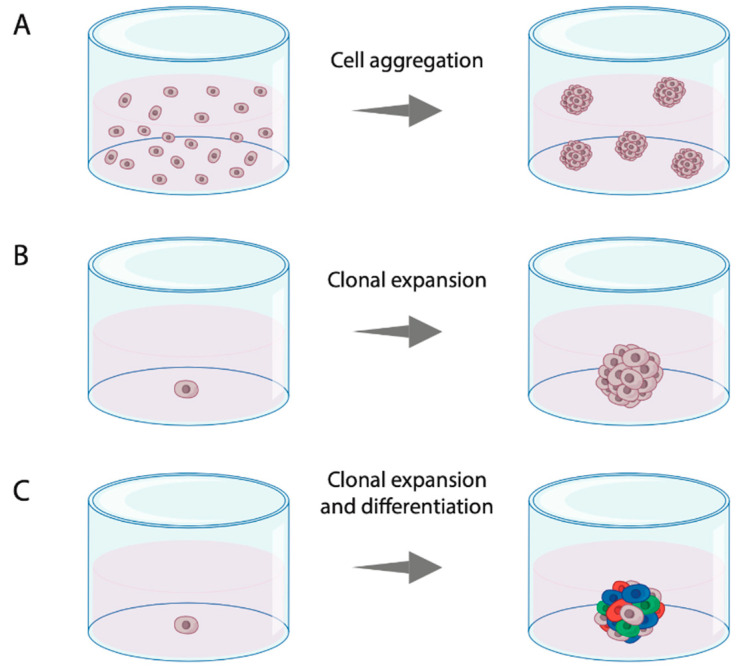
Principles of sphere formation from primary tumor cells. (**A**) Sphere formed by aggregation of multiple cells in a single-cell suspension; (**B**) sphere formed by clonal expansion of a single cell with proliferating potential; (**C**) sphere formed by clonal expansion and lineage-dependent differentiation of a single cell with proliferating potential.

**Table 1 cells-09-02527-t001:** Criteria and scoring for the selection of key publications.

Criterium	Scoring
Reproducibility of methods	0, 1, or 2 points
Number of patients included	0–9: 0 points, 10–29: 0.5 points, ≥30: 1 point
Success percentage	Not reported: 0 points, reported: 1 point
Culture duration	Not reported: 0 points, reported: 1 point
Complete results on culture quality and treatment response	0, 1, or 2 points

**Table 2 cells-09-02527-t002:** Overview of currently used primary culture models for HNSCC.

Culture Model	Description	Examples	Advantages ^a^	Disadvantages ^a^	Ref
**2D**					
Adherent monolayer	Cells grown as a monolayer attached to a plastic surface	Cell Adhesive Matrix assay FLAVINO assay	Appropriate for most cell types Access to nutrients and oxygen is no limiting factor Standardized protocols Simplicity Low cost High reproducibility	Different cell morphology No natural structure of tumor Limited cell–cell and cell–ECM interactions No gradient in nutrient and oxygen availability as in vivo Not all cell suspensions will grow in monolayer setting	[16,17,18,19]
**3D**					
Multicellular spheroids	Cell aggregates grown from either single-cell cultures or tissue fragments with multiple cell types	Suspension culture Fragment spheroids Scaffold based Agitation based * Hanging-drop culture Magnetic levitation *	Physiological cell–cell and cell–ECM interactions Multiple cell types resembling in vivo situation Gradients of nutrient and oxygen availability High reproducibility Suitable for HTS	Often not uniform in size Simplified architecture Difficult to maintain long term Lack potency for self-renewal and differentiation Possibility of central necrosis	[20,21,22,23]
CSC-enriched spheroids	Spheroids enriched for CSCs or cells with stem cell traits, formed by clonal proliferation.	Suspension culture Hanging-drop culture	Suitable to study CSC-related characteristics Potential for self-renewal and differentiation	Absence of non-tumor cells No histological preservation of original tumor Identification of CSCs from solid tumors remains evasive Possibility of central necrosis	[24,25,26,27]
Organoids	Collection of cell types that develops from stem cells or progenitors and self-organizes in a manner similar to in vivo	Embedded in matrix Air–liquid interface CTOS method	Physiological cell–cell and cell–ECM interactions Gradients of nutrient and oxygen availability Composition and architecture resembling primary tissue Capacity of self-renewal and differentiation Can be cryopreserved and expanded	Often not uniform in size May lack key cell types Hard to reach in vivo maturity Possibility of central necrosis Time consuming High costs for media and growth factors Less suitable for HTS	[28,29,30,31]
Histocultures	Tumor tissue left intact by only mechanical cutting/slicing	Histoculture Drug Response Assay Tumor slices Tumor fragments	Tumor environment as in vivo Maintains tumor heterogeneity, including stromal/immune cells No tissue pre-processing	Relatively much tumor tissue needed for establishment Difficult to maintain long term Not suitable for HTS	[32,33,34,35,36,37]
Patient-derived xenograft (PDX)	Patient-derived cancer cells are injected into immune-deficient mice	Subcutaneous implantation Orthotopic implantation	Maintains tumor microenvironment and heterogeneity Captures complexity of metastatic processes in a living system Intact endocrine system	Mice have deficient immunity Differences in microenvironment between mice and human Time consuming High costs Ethical issues of animal use	[38,39,40]
Microdevices	System that provides a precisely controlled culture environment	Microfluidic systems Tumor-on-a-chip models	Can be combined with any culture technique Allows continuous perfusion with culture medium Tightly controlled culture conditions	Requires external materials (tubes, pumps, connectors) to operate Complex to control High costs Still in early development	[41,42,43]

ECM = extracellular matrix, HTS = high-throughput screening, CTOS = cancer tissue-originated spheroid, CSCs = cancer stem cells, * = not described for HNSCC tissues in included literature of this review, ^a^ = (Dis)advantages are extracted or deduced from references and may not be all encompassing. (Dis)advantages can be different for specific methods/examples.

**Table 3 cells-09-02527-t003:** Overview of selected studies using chemotherapy or radiotherapy on various cultures from HNSCC tissue.

Authors, Year	Culture Technique	Patients (N)	Culture Duration (Days)	Culture Success (%)	Ex Vivo Treatment	Response Read-Out Method	Preservation of Tissue Parameters in Culture	Main Results of Treatment	In-Patient Treatment	Correlation Ex Vivo vs. In Patient
Leong, 2014 [57]	Multicellular Spheroids	3	4–9	-	Cisplatin, 5-FU, Etoposide, RT	FACS	-	Spheroids were more resistant to all treatments than monolayers. Cells with a high ALD expression were resistant to cytotoxic agents.	-	-
Lim, 2011 [24]	CSC-enriched Spheroids	47	14	6%	Cisplatin, 5-FU, Paclitaxel, Docetaxel	MTT	-	Undifferentiated spheroid cells were significantly more resistant to chemotherapeutic agents than differentiated spheroid cells.	-	-
Tanaka, 2018 [30]	Organoids	43	8–30	30.2%	Cisplatin, Docetaxel	Relative organoid area day 1 vs. day 8	Histological patterns, vimentin expression and CD44/ALDH1A1 ratios were similar between organoids and the original tumor.	Cisplatin IC50: 0.92–1.02 µM Docetaxel IC50: 1.46–3.75 nM	-	-
Driehuis, 2019 [31]	Organoids	34	42	60%	Cisplatin, Carboplatin, RT	CellTiter-Glo 3-D Assay	Tumor-specificHistopathologic changes were retained in culture. Organoids contain only transformed tumor cells.	IC50 cisplatin: 0.5–12.8 μM IC50 carboplatin: 3.0–81.9 μM AUC RT: 238–698	RT	6/7 matched response: 3 positive outcomes with sensitive organoid, 3 no response with non-sensitive organoid
Au, 1993 [58]	Histocultures	83	9	59%	Cisplatin, 5-FU, MMC	^3^H-TdR	Most histocultures contained areas of viable and necrotic tissue. Histology of viable regions of the cultures was similar to that of the fresh tumor.	Primary tumors mean IC50: 5-FU: 0.68 ± 0.74 µg/mL Cisplatin: 3.77 ± 2.42 µg/mL MMC: 0.25 ± 0.13 µg/mL 9/47 tumors not sensitive	-	-
Robbins, 1994 [32]	Histocultures (HDRA)	26	3–11	88%	Cisplatin	^3^H-TdR	-	84% reduction in the number of cells incorporating ^3^H-TdR in drug-treated samples compared to control samples is used as the cut off for sensitivity in vitro	Cisplatin	Sensitivity: 71% Specificity: 78% PPV: 83% NPV: 64%
Robbins, 1996 [59]	Histocultures (HDRA)	43	6–9	91%	Cisplatin	^3^H-TdR	-	Sensitivity overall: 1.5µg/mL: 22% 15 µg/mL: 62% 37.5 µg/mL: 83% Factor growth inhibition Untreated lesions: ×2.44 Recurrent tumors: ×5.56	-	-
Welters, 1999 [60]	Histocultures (3 mm^3^)	8	1	-	Cisplatin	32-P labeling	-	Because most of the HNSCC biopsies were too small to perform analyses at several time points, no adduct levels over time could be measured.	Cisplatin	DNA adduct levels partial responder vs. non- responder: Pt-GG: 27.4 vs. 5.1 Pt-AG: 13.2 vs. 2.4
Singh, 2002 [61]	Histocultures (HDRA)	41	2	98%	Cisplatin, 5-FU	MTT	-	number of resistant tumors: 13/41 resistant to 5-FU, 13/41 resistant to cisplatin, 11/41 resistant to both	Cisplatin, 5-FU, RT	2 year CSS sensitive vs. not-sensitive: 5-FU: 85% vs. 64% Cisplatin: 86% vs. 63% 5-FU + cisplatin: 85% vs. 63%
Ariyoshi, 2003 [62]	Histocultures (HDRA)	19	7	100%	Cisplatin, Docetaxe l5-FU, THP, ADM, BLM	MTT	-	Sensitivity rate per drug: Cisplatin: 78.9% Docetaxel: 100% 5-FU: 38.4% THP: 7.7% ADM: 0% BLM: 21.4%	Cisplatin, 5-FU, THP, BLM	Accuracy: 78.9% Sensitivity: 86.7% Specificity: 50% TPR: 86.7% TNR: 50%
Hasegawa, 2006 [63]	Histocultures (HDRA)	49	7	100%	Cisplatin, 5-FU	MTT	-	Cisplatin efficacy rate: 36.7–71.4% 5-FU 120 μg/mL vs. 300 μg/mL efficacy rate: 23.1–57.7% vs. 70.8–75.0%	Cisplatin, 5-FU	Prediction rate: 77.8% Sensitivity: 90.9% Specificity: 57.1% TPR: 76.9% TNR: 80.0%
Hasegawa, 2008 [64]	Histocultures (HDRA)	44	7	82%	Cisplatin, 5-FU	MTT	-	Mean I.I. 5-FU: 36.76% Mean I.I. cisplatin: 35.65% 5-FU sensitivity: 21/44 (58.3%) Cisplatin sensitivity: 21/44 (58.3%)	-	-
Pathak, 2008 [65]	Histocultures (HDRA)	57	8	91%	Cisplatin, 5-FU, MTX	MTT	-	Cisplatin sensitivity: 52% 5-FU sensitivity: 46% MTX sensitivity: 52% Sensitive to one drug: 88%	Cisplatin, 5-FU, MTX, Paclitaxel, Ifosfamide	Accuracy: 74% Sensitivity: 79% Specificity: 71% PPV: 69% NPV: 80%
Gerlach, 2013 [34]	Histocultures (Tissue slices 350 µm)	12	3–6	-	Cisplatin, Docetaxel	IHC	Cultures maintained morphological features and γH2AX expression for up to 6 days compared to original histopathology.	Control vs. cisplatin vs. docetaxel: # nuclei: ±400 vs. ±125 vs. ±150 % caspase-3-positive cells: ±2% vs. ±6% vs. ±22%	-	-
Suzuki, 2015 [66]	Histocultures (HDRA)	28	7	100%	Cisplatin	MTT	-	SUV_max_: 14.04 ± 7.52 I.I.: 50.98 ± 26.6 SUV_max_ was significantly correlated with the I.I. cisplatin (*p* < 0.04, R^2^ = 0.17)	Cisplatin, 5-FU, RT	SUV_max_ ≥ 10.5 and I.I. cisplatin < 50 were significantly correlated with shorter OS
Engelmann, 2020 [49]	Histocultures	13	7–21	100%	RT	IHC	Comparable histological and morphological characteristics were observed between primary non-HPV tumors and histocultures after 14 days. Cultures display heterogeneous growth patterns on dermal equivalent.	Irradiation of tissues resulted in a slight increase or decrease in Ki-67 expression compared to control: Overall: +0.22% Non-HPV driven: −5.28% HPV driven: +3.89% 2/5 tumors showed increase in apoptotic cells after fractionated irradiation.	RT	One patient developed local relapse, with the corresponding histoculture showing an invasive growth pattern
Hattersley, 2012 [41]	Microdevice	23	8	91%	Cisplatin, 5-FU	LDH and cytochrome-c release, WST-1 metabolism	The nuclei of the tissue 72 h after culture appear intact and loss of cell cohesion is minimal. There was no necrosis in the center of the biopsy.	% viable cells after treatment: Control: 72% ± 15.6 5-FU: 45% ± 22.3 Cisplatin: 44% ± 20.2 5-FU + cisplatin: 30% ± 23.7 All treatments showed a higher release of cytochrome-c than control samples. (*p* < 0.01)	-	-
Carr, 2014 [67]	Microdevice	35	2–3	-	RT	LDH and cytochrome-c release, IHC	There was no difference between the apoptotic index (AI) of the uncultured and cultured control tissue (*p* = 0.29).	AI 0 Gy: ±1% AI 5 Gy: ±7% AI 10 Gy: ±15% AI 20 Gy: ±20% AI 40 Gy: ±45% (*p* = 0.006)	-	-
Cheah, 2017 [68]	Microdevice	5	2	100%	Cisplatin, RT	LDH release, IHC, TUNEL	-	γH2AX: 1/5 sign. response CK18-LI: 2/5 sign. responseT UNEL: 3/4 sign. response Ki-67: 0/5 sign. response	RT, CRT	Matched responses for 2/2 patients (for 2/4 markers)
Kennedy, 2019 [69]	Microdevice	18	3	67%	Cisplatin, RT	IHC	The average Ki-67 index decreased in the control sample (7.9% ± 3.5) relative to the pre-culture sample.No difference in γH2AX expression and apoptosis between pre-culture and control samples.	Control vs. RT: BrdU: 13.3% vs. 7.0%, Ki-67: 15.3% vs. 4.0%, γH2AX: 76.6% vs. ±90%, Caspase cleaved cytokeratin 18: ±3% vs. ±12% Addition of cisplatin: 1.9-fold increase in apoptotic index	-	-

5-FU = 5-fluoroucil, RT = radiotherapy, FACS = fluorescence-activated cell sorting, ALD = aldehyde dehydrogenase, CSC = cancer stem cell, MTT = 3-(4,5-dimethylthiazol-2-yl)-2,5-diphenyltetrazolium bromide, ALDH1A1 = aldehyde dehydrogenase 1 family, member A1, IC50 = half maximal inhibitory concentration, MMC = mitomycin C, ^3^H-TdR = [^3^H] radiolabeled thymidine, HDRA = Histoculture Drug Response Assay, PPV = positive predictive value, NPV = negative predictive value, pt-DNA = platinum-DNA, CSS = cause-specific survival, THP: 4-0-tetrahydropyranyl adriamycin, ADM = adriamycin, BLM = bleomycin, TPR = true positive ratio, TNR = true negative ratio, I.I = Inhibition Index, MTX = methotrexate, IHC = immunohistochemistry, SUV = standardized uptake value, OS = overall survival, LDH = lactate dehydrogenase, WST-1 = 4-[3-(4-iodophenyl)-2-(4-nitro-phenyl)-2H-5-tetrazolio]-1,3-benzene sulfonate, AI = apoptotic index, Gy = Gray, TUNEL = terminal deoxynucleotidyl transferase dUTP nick end labeling, and CRT = chemoradiotherapy, # = number.

**Table 4 cells-09-02527-t004:** Overview of selected studies using immunotherapy or targeted therapy on various cultures from HNSCC tissue.

Authors, Year	Culture Technique	Patients (N)	Culture Duration (Days)	Culture Success (%)	Ex Vivo Treatment	Response Read-Out Method	Preservation of Tissue Parameters in Culture	Main Results of Treatment
Kross, 2007 [70]	Multicellular spheroids	18	14	>90%	LLME	ELISA, LIVE/DEAD kit, BrdU labeling	Nearly 100% of the spheroid surface consisted of live cells, indicating viability after 14 days of culture in vitro.	Mean IL-6 production in 168 h, control vs. treated: ±17.500 vs. ±5000 pg/mL Mean MCP-1 production in 168 h, control vs. treated: ± 7000 vs. ± 1000 pg/mL
Kloss, 2015 [71]	Multicellular spheroids	5	11–14	100%	Cetuximab	Cytometric bead array, fluorescent, microscopy, FACS	-	When cetuximab was absent, the NK cells showed clearly impaired and disordered “effector-to-target” interactions and decreased both cancer cell cluster infiltrations and cancer cell killing.
Sun, 2014 [72]	CSC-enriched spheroids	3	6	-	c-Met inhibitor PF-2341066	Sphere-forming ability	Immunofluorescent staining showed that the spheres have high expression levels of several known CSC markers.	Sphere formation was inhibited in a dose-dependent manner. CSC cells were more sensitive to PF-2341066 than to docetaxel. In contrast, differentiated cells show the opposite effect
Driehuis, 2019 [31]	Organoids	34	42	60%	Nutlin-3 Cetuximab, Alpelisib, Vemurafenib, Everolimus, AZD4547 Niraparib	CellTiter-Glo 3-D Assay	Tumor-specific histopathologic changes were retained in culture.Organoids contain only transformed tumor cells.	IC50 nutlin-3: 0.5–22.6 µM AUC cetuximab: 93.94–180.7 IC50 alpelisib: 0.12–4.12 µM IC50 everolimus: 0.00–19.83 µM IC50 AZD4547: 0.67–28.38 µM IC50 niraparib: 4.24–25.66 µM
Dean, 2010 [73]	Histocultures (800–1000 µm)	22	3	86.4%	Anti-EMMPRIN mAb, Cetuximab	ATP viability assay, TUNEL	Cultures had excellent viability over 72 h. Less than 5% of any specimen showed necrosis.	Average ATP level anti-EMMPRIN vs. cetuximab: 57% vs. 45% (control: 100%) (*p* = 0.13)Apoptosis was increased in anti-EMMPRIN-treated cultures (77%) vs. controls (30%).
Gerlach, 2013 [34]	Histocultures (350 µm)	12	3–6	-	Cetuximab	LDH release, IHC, TUNEL	Slice cultures maintained morphological features for up to 6 days as compared to the original diagnostic histopathology. No change of γH2AX positivity was visible at any of the tested time points.	# nuclei control vs. cetuximab: ±400 vs. ±25 Percentage of caspase-3-positive cells control vs. cetuximab: ±2% vs. ±5%
Freudl-sperger, 2014 [35]	Histocultures (300–350 μm)	15	6	-	LY294002	IHC	Histological staining confirmed preservation of tissue architecture. The cultures showed almost 100% Ki-67 staining and few apoptotic cells.	Expression after treatment with LY294002 vs. RT vs. LY294002 + RT (control 100%): p-AKT: ±65% vs. ±135% vs. ±55% p-H2AX: ±80% vs. ±900% vs. ±1700% Ki-67: ±80% vs. ±70% vs. ±35%
Peria, 2015 [36]	Histocultures (300 μm)	5	3	80%	Cetuximab, Sorafenib	IHC	After 72 h, an increase in necrosis was observed in cultured tumor slices. After 48 h, proliferation decreased by 30–70%.	Average % Ki-67-positive cells, control vs. cetuximab vs. sorafenib: ±25% vs. ±15% vs. ±21%
Rauth, 2016 [74]	Histocultures (2–3 mm^3^)	5	3	100%	Lupeol	IHC	Key components of tumor microenvironment were found to be intact up to 3 days.	Tumor cell content control vs. Lupeol: ±70% vs. ±45% (*p* <0.05) Ki-67-positive cells control vs. Lupeol: ±15% vs. ±2% (*p* < 0.01)
Affolter, 2016 [75]	Histocultures (800–1000 µm)	9	6	100%	MEK inhibitor PD-0325901	IHC	The number of Ki-67-positive tumor cells was 5% to 7.5% in non-treated cultures. In 1 culture, 75% of all cells were positive for Ki-67 in the control. γH2AX expression levels varied widely between 10% and 95%.	Expression after treatment with 0 μM PD-0325901 + 5 Gy vs. 20 μM PD-0325901 + 5 Gy: pERK: 27.8% vs. 4.4% Ki-67: 8.1% vs. 1.8% γH2AX: 43.1% vs. 43.1%
Donna-dieu, 2016 [76]	Histocultures (300 μm)	18	2	78%	Rapamycin Sorafenib Cetuximab Erlotinib Masatinib Ponatinib Afatinib Tivantinib	IHC	-	Average % of cell inhibition (control 100%): Rapamycin: 77.1% Sorafenib: 65.7% Cetuximab: 73.4% Erlotinib: 75.9% Masatinib: 70.5% Ponatinib: 74.2% Afatinib: 60.9% Tivantinib: 80.9%
Al-Samadi, 2019 [42]	Microdevice	5	3	-	IDO 1 inhibitor, PD-L1 antibody	Fluorescent microscopy-based cell counting	-	AUC # of infiltrated immune cells Control vs. IDO 1 vs. PDL-L1: Patient 4: ±550 vs. ±850 vs. ±400 Patient 5: ±0 vs. ±250 vs. ±0 AUC cancer cell proliferation rate: Patient 4: ±1.0 vs. ±0.85 vs. ±0.4 Patient 5: ±1.0 vs. ±0.7 vs. ±0.8

LLME = L-leucine-methylester, ELISA = enzyme-linked immunosorbent assay, BrdU = bromodeoxyuridine, F-spheroids = fragment spheroids, IL-6 = interleukin-6, MCP-1 = monocyte chemoattractant protein-1, FACS = fluorescence-activated Cell Sorting, NK cells = Natural Killer cells, CSC = cancer stem cell, AUC = area under the curve, EMMPRIN = extracellular matrix metalloproteinase inducer, mAb = monoclonal antibody, ATP = adenosine triphosphate, TUNEL = terminal deoxynucleotidyl transferase dUTP nick end labeling, LDH = lactate dehydrogenase, IHC = immunohistochemistry, RT = radiotherapy, and Gy = Gray; # = number.

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
