# Peer review of "Ex Vivo Culture Models to Indicate Therapy Response in Head and Neck Squamous Cell Carcinoma"

_cells, 2020, doi:10.3390/cells9112527_

Round 1

Reviewer 1 Report

I have made some suggestions in the attached file

Reviewer 2 Report

This review article provides an in-depth and critical analysis of the available literature about the response to therapy prediction power of ex-vivo HNSCC culture. It is clearly organized, very well written and very significant to the field. Besides minor typos, I suggest the manuscript to be accepted for publication, and warmly congratulate the author for the quality of their work.

Author Response

We would like to thank this reviewer for the effort of reviewing this manuscript carefully and for the kind words.

Reviewer 3 Report

I have no edits to suggest.

The review is well-written, organized, and covers a timely topic very comprehensively. I think that the terminology was accurate, although perhaps not quite so accurate in the literature out there.

The authors are to be congratulated for such a thorough review.

Author Response

(The authors gave the same response as above.)
